# LM-HT SNN: Enhancing the Performance of SNN to ANN Counterpart through Learnable Multi-hierarchical Threshold Model

**Zecheng Hao**[1], **Xinyu Shi**[1,2], **Yujia Liu**[1], **Zhaofei Yu**[1,2*] **& Tiejun Huang**[1,2]
[1] School of Computer Science, Peking University
[2] Institute for Artificial Intelligence, Peking University

## Abstract

Compared to traditional Artificial Neural Network (ANN), Spiking Neural Network (SNN) has garnered widespread academic interest for its intrinsic ability to transmit information in a more energy-efficient manner. However, despite previous efforts to optimize the learning algorithm of SNNs through various methods, SNNs still lag behind ANNs in terms of performance. The recently proposed multi-threshold model provides more possibilities for further enhancing the learning capability of SNNs. In this paper, we rigorously analyze the relationship among the multi-threshold model, vanilla spiking model and quantized ANNs from a mathematical perspective, then propose a novel LM-HT model, which is an equidistant multi-threshold model that can dynamically regulate the global input current and membrane potential leakage on the time dimension. The LM-HT model can also be transformed into a vanilla single threshold model through reparameterization, thereby achieving more flexible hardware deployment. In addition, we note that the LM-HT model can seamlessly integrate with ANN-SNN Conversion framework under special initialization. This novel hybrid learning framework can effectively improve the relatively poor performance of converted SNNs under low time latency. Extensive experimental results have demonstrated that our model can outperform previous state-of-the-art works on various types of datasets, which promote SNNs to achieve a brand-new level of performance comparable to quantized ANNs. Code is available at `https://github.com/hzc1208/LMHT_SNN`.

## 1 Introduction

Recognized as the third generation of artificial neural networks [33], Spiking Neural Network (SNN) is increasingly receiving significant academic attention due to its enormous potential in biological plausibility and high energy efficiency. As the information transmission between the pre-synaptic and post-synaptic layers relies on the discrete spike signal, which will be only emitted when the membrane potential of the corresponding neuron exceeds the firing threshold, SNNs have a unique event-driven property compared to conventional Artificial Neural Network (ANN). By utilizing this property, researchers have pointed out that SNNs can achieve significant advantages in terms of energy consumption on neuromorphic hardware [35, 5, 37]. Currently, SNNs have further fulfilled a role in multiple application scenarios including object detection [22], natural language processing [32], and 3D recognition [24].

Spatial-Temporal back-propagation (STBP) with surrogate gradients is currently the most mainstream supervised learning algorithm suitable for SNNs. Although previous works have attempted to further enhance the learning ability of SNNs by delving into various optimization strategies, including

---

*Corresponding author: yuzf12@pku.edu.cn

gradient adjustment [29, 8, 14] and structural improvement [54, 51, 44, 50], there is still a certain performance gap between ANNs and SNNs.

Recently, the STBP learning algorithm based on multi-threshold models [41, 42] is considered as another potential way to improve the performance of SNNs. In this scenario, multiple levels of the firing threshold enable SNNs to transmit richer information at each time-step. Unfortunately, we think that current related works have not accurately recognized the mathematical essence of multi-threshold models as well as their relationship with ANNs and SNNs. In this paper, we innovatively propose a learnable multi-hierarchical and equidistant threshold model based on global input information, which is called LM-HT model. On the one hand, we note that our LM-HT model can equivalently represent the information of the vanilla model over multiple consecutive time-steps within a single step. Furthermore, we can convert the LM-HT model into a vanilla single threshold model through a layer-by-layer reparameterization scheme. On the other hand, the STBP method based on the LM-HT model can be transformed into the training modes of the vanilla STBP and quantized ANNs under different parameter initialization conditions, respectively. The main contribution of this work has been summarized as follows:

- We point out that the essence of the equidistant multi-threshold model is to simulate the spike firing situation of the vanilla spiking model within specific time windows. Specially, when the input current follows a completely uniform distribution on the time dimension, its spike firing rate is mathematically equivalent to the activation output of quantized ANNs.

- We propose an advanced LM-HT model, which can enhance the performance of SNNs to the level of ANNs and be transformed into a vanilla single threshold model losslessly during the inference stage. By adopting different parameter initialization schemes, the LM-HT model can further establish a bridge between the vanilla STBP and quantized ANNs training.

- We further design a brand-new hybrid training framework based on the LM-HT model, which is enable to effectively improve the performance degradation problem of traditional ANN-SNN Conversion methods regardless of the time latency degree involved.

- Experimental results have indicated that our model can fulfill state-of-the-art learning performance for various types of datasets. For instance, we achieve the top-1 accuracy of 81.76% for CIFAR-100, ResNet-19 within merely 2 time-steps.

## 2   Related Works

**STBP supervised training.** STBP is the most prevailing recurrent-mode learning algorithm in the field of SNN direct training. Wu *et al.* [47] tackled the non-differentiable problem existed in the spike firing process by utilizing surrogate gradients and achieved gradient smoothing calculation between layers. Deng *et al.* [8] and Guo *et al.* [14] respectively proposed brand-new target loss functions by analyzing the temporal distribution of the spike sequence and membrane potential. Furthermore, various temporal-dependent batch normalization layers [54, 10, 15] and advanced spiking neuron models [51, 44] have been pointed out, which enhances the capability and stability of SNN learning. The researchers also designed a variety of residual blocks [11, 20] and Transformer structures [55, 49] suitable for SNNs, promoting the development of STBP training towards the domains of deep and large-scale models. In addition, some variant and extended learning methods based on STBP have also received widespread attention. Temporal Coding [36] and Time-to-First-Spike (TTFS) [21] algorithm conduct one-time back-propagation based on the specific firing moment. Meng *et al.* [34] introduced the idea of online learning into vanilla STBP algorithm, which significantly saves training memory overhead by eliminating the gradient chains between different time-steps. Fang *et al.* [12] proposed a spiking neuron model that supports parallel computing in forward propagation, which also provides inspiration for this work.

**ANN-SNN Conversion.** ANN-SNN Conversion is another widely used method for obtaining high-performance SNNs with limited computational resources, which establishes a mathematical mapping relationship between activation layers and the Integrate-and-Fire (IF) models. Cao *et al.* [3] first proposed a two-step conversion learning framework, which replaces the activation functions of pre-trained ANNs with the IF models layer by layer. On this basis, Han *et al.* [16] and Li *et al.* [28] classified and summarized the relevant errors existed in the conversion process. Deng *et al.* [7] and Bu *et al.* [2] further reduced the conversion errors through deriving the optimal values for the bias term and initial membrane potential. For the critical conversion error caused by uneven spike firing

sequences, multiple optimization strategies have been proposed successively, including memorizing the residual membrane potential [17], firing negative spikes [43, 25], calibrating offset spikes [18] and hybrid finetuning training [45]. Currently, ANN-SNN Conversion has been further applied to the training of large-scale visual and language models [46, 32].

**Spiking neural models with multi-threshold.** The current proposed multi-threshold models can be generally divided into two categories: one emits signed spikes [22, 52, 43], while the other emits multi-bit spikes [27, 41, 42, 24]. However, these works generally consider using multi-threshold models to reduce ANN-SNN Conversion errors and lack further theoretical analysis. In this paper, we have the foresight to recognize the mathematical equivalence relationship between equidistant multi-threshold models and quantized ANNs under the conditions of using the soft-reset mechanism and uniform input current, achieving the current optimal performance in the domain of STBP learning.

## 3 Preliminaries

**The spiking neuron models for SNNs.** The Leaky-Integrate-and-Fire (LIF) model is one of the most commonly used models in the current SNN community. The following equations have depicted the dynamic procedure of the LIF model in a discrete form:

$$\boldsymbol{m}_{LIF}^l(t) = \lambda_{LIF}^l \boldsymbol{v}_{LIF}^l(t-1) + \boldsymbol{I}^l(t), \ \boldsymbol{I}^l(t) = \boldsymbol{W}^l \boldsymbol{s}_{LIF}^{l-1}(t)\theta^{l-1}. \tag{1}$$

$$\boldsymbol{v}_{LIF}^l(t) = \boldsymbol{m}_{LIF}^l(t) - \boldsymbol{s}_{LIF}^l(t)\theta^l, \ \boldsymbol{s}_{LIF}^l(t) = \begin{cases} 1, & \boldsymbol{m}_{LIF}^l(t) \geq \theta^l \\ 0, & \text{otherwise} \end{cases}. \tag{2}$$

Eq.(1) describes the charging process: $\forall t \in [1, T]$, $\boldsymbol{m}_{LIF}^l(t)$ and $\boldsymbol{v}_{LIF}^l(t-1)$ respectively represent the membrane potential before and after the charging at the $t$-th time-step. $\boldsymbol{I}^l(t)$ denotes the input current and $\lambda_{LIF}^l$ characterizes the leakage degree of the membrane potential. When $\lambda_{LIF}^l = 1$, the LIF model will degenerate into a more specialized model called the IF model. Eq.(2) depicts the reset and firing process: $\boldsymbol{s}_{LIF}^l(t)$ indicates the spike emitting situation and $\theta^l$ is the firing threshold. Here we adopt the soft-reset mechanism, which means that the reset amplitude of the membrane potential is equal to the value of $\theta^l$.

**STBP learning algorithm for SNNs.** The gradient calculation mode of STBP is inspired by the back-propagation Through Time (BPTT) algorithm in Recurrent Neural Network (RNN), which will propagate along the spatial and temporal dimensions of SNNs simultaneously. Following equations have described the specific propagation process:

$$\frac{\partial \mathcal{L}}{\partial \boldsymbol{m}_{LIF}^l(t-1)} = \underbrace{\frac{\partial \mathcal{L}}{\partial \boldsymbol{s}_{LIF}^l(t-1)}\frac{\partial \boldsymbol{s}_{LIF}^l(t-1)}{\partial \boldsymbol{m}_{LIF}^l(t-1)}}_{\text{spatial dimension}} + \underbrace{\frac{\partial \mathcal{L}}{\partial \boldsymbol{m}_{LIF}^l(t)}\frac{\partial \boldsymbol{m}_{LIF}^l(t)}{\partial \boldsymbol{v}_{LIF}^l(t-1)}\frac{\partial \boldsymbol{v}_{LIF}^l(t-1)}{\partial \boldsymbol{m}_{LIF}^l(t-1)}}_{\text{temporal dimension}}, \tag{3}$$

Here $\mathcal{L}$ denotes the target loss function. From Eq.(2) one can note that the mathematical relationship between $\boldsymbol{s}_{LIF}^l(t)$ and $\boldsymbol{m}_{LIF}^l(t)$ is equivalent to $\boldsymbol{s}_{LIF}^l(t) = H(\boldsymbol{m}_{LIF}^l(t) - \theta^l)$, where $H(\cdot)$ denotes Heaviside step function. As Heaviside function is non-differentiable, researchers consider using a surrogate function, which is approximate to Heaviside function but differentiable, to handle the term $\frac{\partial \boldsymbol{s}_{LIF}^l(t)}{\partial \boldsymbol{m}_{LIF}^l(t)}$ in the back-propagation chain. For example, $\frac{\partial \boldsymbol{s}_{LIF}^l(t)}{\partial \boldsymbol{m}_{LIF}^l(t)} = \text{sign}\left(\left|\boldsymbol{m}_{LIF}^l(t) - \theta^l\right| \leq \frac{\theta^l}{2}\right)$ describes the well-known rectangular surrogate function.

**Quantized ANNs.** The quantized ANN model is a widely used structure in the field of ANN-SNN Conversion. Compared to traditional ANNs, quantized ANNs usually use the following Quantization-Clip-Floor-Shift (QCFS) function [28, 2] as their activation function:

$$\boldsymbol{a}^l = \frac{\vartheta^l}{T_q}\text{clip}\left(\left\lfloor \frac{\boldsymbol{W}^l \boldsymbol{a}^{l-1}T_q + \varphi^l}{\vartheta^l} \right\rfloor, 0, T_q\right). \tag{4}$$

Here $\boldsymbol{a}^l$ and $\varphi^l$ represent the activation output and shift factor, while $T_q$ and $\vartheta^l$ denote the quantization level and learnable scaling factor. If we set $T_q = T, \vartheta^l = \theta^l, \boldsymbol{a}^l = \sum_{t=1}^T \boldsymbol{s}_{IF}^l(t)\theta^l/T, \boldsymbol{v}_{IF}^l(0) = \varphi^l$, one can find that the so-called QCFS function actually simulate the average spike firing rate of the IF model (we set $\boldsymbol{r}_{IF}^l(T_q) = \sum_{t=1}^{T_q} \boldsymbol{s}_{IF}^l(t)\theta^l/T_q$) under the condition of the uniform input current and soft-reset mechanism. This conclusion suggests that SNNs have the potential to maintain the same level of performance as ANNs under specific conditions.

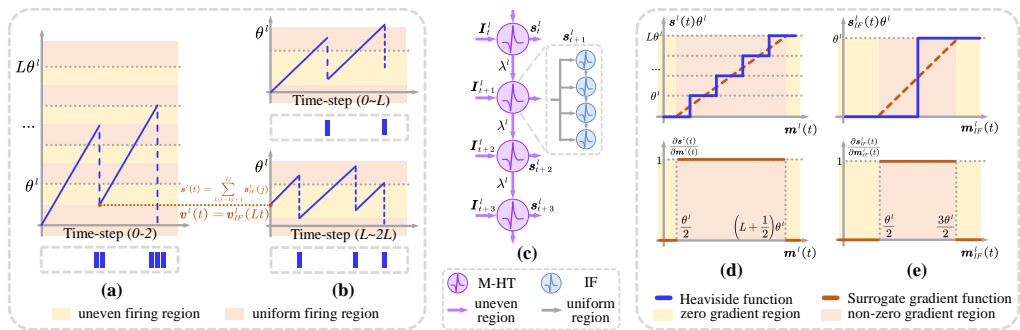

Figure 1: Forward and backward propagation of the M-HT model. (a)-(c): mathematical relationship between the M-HT model and vanilla IF model. (d)-(e): surrogate gradient calculation for the M-HT model.

# 4 Methodology

## 4.1 The Multi-hierarchical Threshold (M-HT) Model

In this section, we first introduce the M-HT model, which has equidistant multi-level thresholds and will select the threshold closest to its current membrane potential at each time-step to achieve the process of firing spikes and resetting potential. Eqs. (5)-(6) describe the dynamic equations of the M-HT model.

$$\boldsymbol{m}^l(t) = \lambda^l \boldsymbol{v}^l(t-1) + \boldsymbol{I}^l(t), \ \boldsymbol{I}^l(t) = \boldsymbol{W}^l \boldsymbol{s}^{l-1}(t)\theta^{l-1}. \tag{5}$$

$$\boldsymbol{v}^l(t) = \boldsymbol{m}^l(t) - \boldsymbol{s}^l(t)\theta^l, \ \boldsymbol{s}^l(t) = \begin{cases} L, & \boldsymbol{m}^l(t) \geq L\theta^l \\ k, & k\theta^l \leq \boldsymbol{m}^l(t) < (k+1)\theta^l, k = 1,...,L-1 \\ 0, & \text{otherwise} \end{cases} \tag{6}$$

Here $L$ denotes the number of level for the firing threshold. Regarding the surrogate gradient calculation of the M-HT model, similar to the vanilla spiking models, we propose $\frac{\partial \boldsymbol{s}^l(t)}{\partial \boldsymbol{m}^l(t)} = \text{sign}\left(\frac{1}{2}\theta^l \leq \boldsymbol{m}^l(t) \leq (L+\frac{1}{2})\theta^l\right)$, which covers a wider range of the membrane potential, as shown in Fig.1(d)-(e). As the M-HT model has $L$ different firing options at each time-step, we can consider the information transmitted by the M-HT model within one time-step as an information integration of the vanilla model for $L$ time-steps. Therefore, we attempt to bridge a mathematical equivalent relationship between the M-HT and IF model:

**Lemma 4.1.** $\forall t \in [1, T]$, if $\boldsymbol{v}^l(t-1) \in [0, \theta^l]$, the effect of inputting current $\boldsymbol{I}^l(t)$ into a M-HT model with $L$-level threshold at the $t$-th time-step, is equivalent to continuously inputting uniform current $\boldsymbol{I}^l(t)/L$ for $L$ time-steps into a IF model with $\boldsymbol{v}^l_{IF}(0) = \boldsymbol{v}^l(t-1)$, i.e. $\boldsymbol{s}^l(t) = clip\left(\left\lfloor \frac{\boldsymbol{v}^l(t-1)+\boldsymbol{I}^l(t)}{\theta^l} \right\rfloor, 0, L\right) = \sum_{j=1}^{L} \boldsymbol{s}^l_{IF}(j)$.

Lemma 4.1 indicates that the M-HT model under a single time-step can be used to simulate the total number of spikes emitted by the IF model under uniform input current within $L$ consecutive time-steps. In addition, note that $\boldsymbol{s}^l(t)$ in Lemma 4.1 can also be calculated through $clip\left(\lfloor \cdot \rfloor, \cdot, \cdot\right)$, which is equivalent to the QCFS function mentioned before in quantized ANNs. The above conclusion preliminarily demonstrates that the M-HT model can achieve the same-level performance as pre-trained ANNs with $L$-level quantization under single-step condition.

## 4.2 The Representation Ability of the M-HT Model on Multiple Time-steps

Based on Lemma 4.1, we further consider the information representation of the M-HT model on multiple time-steps:

**Theorem 4.2.** When $\lambda^l = 1, \boldsymbol{v}^l(0) \in [0, \theta^l)$, for a M-HT model with $L$-level threshold, after $T$ time-steps, we will derive the following conclusions:

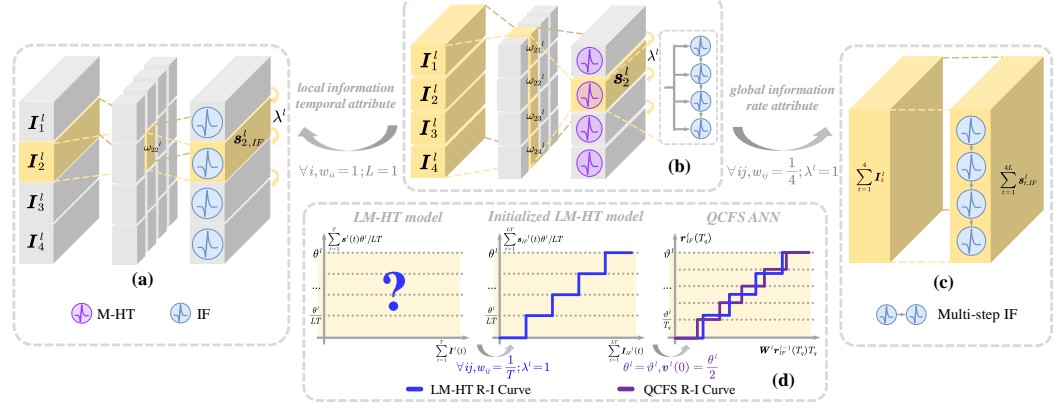

Figure 2: The STBP learning framework based on the LM-HT model. (a): vanilla STBP training. (b): STBP training with the LM-HT model. (c): direct training of quantized ANNs. (d): hybrid training with the LM-HT model, here R-I Curve denotes Rate-Input Curve.

*(i) If we further assume* $\forall t \in [1,T], \boldsymbol{I}^l(t) \in [0, L\theta^l)$, *we will have:* $\forall t \in [1,T], \boldsymbol{s}^l(t) = \sum_{j=L(t-1)+1}^{Lt} \boldsymbol{s}_{IF}^l(j), \boldsymbol{v}^l(t) = \boldsymbol{v}_{IF}^l(Lt), \sum_{t=1}^{T} \boldsymbol{s}^l(t) = \sum_{j=1}^{LT} \boldsymbol{s}_{IF}^l(j)$.

*(ii) If we further assume* $\boldsymbol{I}^l(1) = ... = \boldsymbol{I}^l(T)$, *we will have:* $\sum_{t=1}^{T} \boldsymbol{s}^l(t) = clip\left(\left\lfloor \frac{\boldsymbol{v}^l(0) + \sum_{t=1}^{T} \boldsymbol{I}^l(t)}{\theta^l} \right\rfloor, 0, LT\right)$.

*Here the IF model has uniform input currents* $\boldsymbol{I}^l(1)/L, ..., \boldsymbol{I}^l(T)/L$ *respectively within every* $L$ *steps and satisfies* $\boldsymbol{v}_{IF}^l(0) = \boldsymbol{v}^l(0)$.

The proofs of Lemma 4.1 and Theorem 4.2 have been provided in the Appendix. From Theorem 4.2(i) and Fig.1(a)-(c), one can find that the M-HT model is actually equivalent to dividing the spike firing sequence of the IF model on consecutive $LT$ steps into $T$ $L$-step time windows. Combining with the soft-reset mechanism, the M-HT model actually focuses on a specific time window of the vanilla IF model at each time-step and maintains an equal membrane potential with the IF model at the end of each time window (*i.e.* $\forall t \in [1,T], \boldsymbol{v}^l(t) = \boldsymbol{v}_{IF}^l(Lt)$). The M-HT model follows the assumption of uniform input current within each window, while maintaining the basic calculation properties of spiking neurons between different windows. When the input current follows a complete uniform distribution, according to Theorem 4.2(ii), the M-HT model can further simulate the output of an ANN with $LT$-level quantization.

### 4.3 The Learnable Multi-hierarchical Threshold (LM-HT) Model

**The uniform and uneven firing regions in the M-HT model.** For a specific spike firing rate, the M-HT model can often provide multiple spike firing sequences. For example, $[1,1], [0,2], [2,0]$ can all represent the situation where 2 spikes are emitted within 2 time-steps, while only $[1,1]$ can be viewed as a case of uniform firing situation. However, even when the input current is uniformly distributed, as the sum of spikes that cannot be divided by $L$ in $[0, LT]$ is unable to be represented by a uniform spike output sequence, there are still uneven firing situations:

**Corollary 4.3.** *If* $\lambda^l = 1, \boldsymbol{v}^l(0) = 0$ *and* $\boldsymbol{I}^l(1) = ... = \boldsymbol{I}^l(T)$, *for a M-HT model with L-level threshold,* $\boldsymbol{s}^l(1) = ... = \boldsymbol{s}^l(T)$ *is only satisfied when* $\boldsymbol{I}^l(1) \in [k\theta^l, k\theta^l + \theta^l/T), \forall k = 0, ..., L-1$ *or* $\boldsymbol{I}^l(1) \in (-\infty, 0) \cup [L\theta^l, +\infty)$.

The proof is provided in the Appendix. From Corollary 4.3, we can divide the input current into uniform and uneven firing regions according to the corresponding intervals, as shown in Fig.1(a)-(b). Note that the uneven spike sequences emitted by the $l$-th layer may further cause the input current of the $l+1$-th layer to no longer follow the uniform distribution. That is to say, as the number of layers increases, the uneven firing cases will tend to increase gradually without introducing extra regulation.

**Learnable Temporal-Global Information Matrix and leaky parameters.** Enhancing the uniform firing pattern can promote SNNs to achieve superior performance similar to quantized ANNs, while

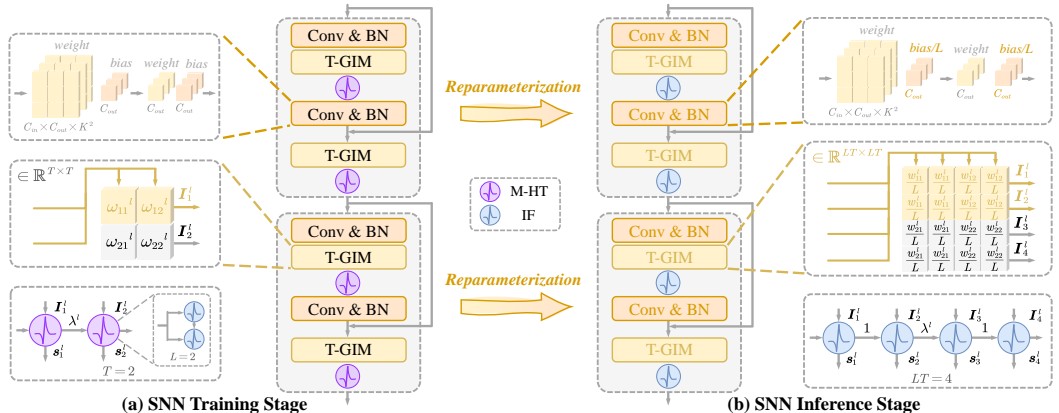

Figure 3: Reparameterization procedure of the LM-HT model.

uneven spike sequences retain more temporal and biological characteristics. Therefore, how to comprehensively utilize these two spike firing patterns becomes a critical problem. To address this issue, we first introduce the concept of Temporal-Global Information Matrix (T-GIM):

$$\forall t \in [1, T], \boldsymbol{I}^l(t) = \sum_{j=1}^{T} \omega_{tj}^l \boldsymbol{W}^l \boldsymbol{s}^{l-1}(j)\theta^{l-1}. \tag{7}$$

Here $\omega_{tj}^l$ is the element at row $t$ and column $j$ of the T-GIM $\boldsymbol{\Omega}^l$, $\boldsymbol{\Omega}^l \in \mathbb{R}^{T \times T}$. As shown in Eq.(7) and Fig.2(b), this brand-new input current adopts a multi-step current weighting form, allowing the model to simultaneously focus on the global information along the time dimension. Note that the new input current will follow a uniform distribution when $\forall i, j \in [1, T], \omega_{ij}^l = \frac{1}{T}$ and degrade to the vanilla input current when $\boldsymbol{\Omega}^l = \mathrm{diag}(1, ..., 1)$. For the first case mentioned above, if we further add the condition $\lambda^l = 1$, according to Theorem 4.2(ii), one can find that the output of the model will be consistent with the activation output of a $LT$-level quantized ANN layer by layer, as shown in Fig.2(b)-(c). For the second case, when $L = 1$, the model will degenerate into vanilla LIF model, as shown in Fig.2(a)-(b).

To enable the model to dynamically adjust the above calculation process, we set both $\boldsymbol{\Omega}^l$ and $\lambda^l$ as learnable parameters. The initial values of $\boldsymbol{\Omega}^l$ and $\lambda^l$ are set to $1/T$ and $1$, respectively. During the training process, we choose the Sigmoid function $\sigma(\cdot)$ to control the parameters for fulfilling smooth gradient updates within a bounded learning range. We call this novel model as Learnable Multi-hierarchical Threshold (LM-HT) Model, which combines T-GIM and learnable attributes. We think the LM-HT model can regulate its spike firing pattern more flexibly and reasonably.

Since we can regulate the computational relationships between different time-steps through learnable $\boldsymbol{\Omega}^l$ and $\lambda^l$ in the LM-HT model, during the back-propagation process, unlike Eq.(3), we detach the term $\frac{\partial \mathcal{L}}{\partial \boldsymbol{m}^l(t)} \frac{\partial \boldsymbol{m}^l(t)}{\partial \boldsymbol{v}^l(t-1)} \frac{\partial \boldsymbol{v}^l(t-1)}{\partial \boldsymbol{m}^l(t-1)}$ from the gradient calculation graph, thereby reducing redundant calculations and completely leaving the gradient propagation between different time-steps to $\boldsymbol{\Omega}^l$ and $\lambda^l$ for control. The back-propagation calculation chains for the LM-HT model have been described as follow. Here $\odot$ denotes the Hadamard product.

$$\frac{\partial \mathcal{L}}{\partial \boldsymbol{m}^l(t)} = \frac{\partial \mathcal{L}}{\partial \boldsymbol{s}^l(t)} \frac{\partial \boldsymbol{s}^l(t)}{\partial \boldsymbol{m}^l(t)}, \quad \frac{\partial \boldsymbol{s}^l(t)}{\partial \boldsymbol{m}^l(t)} = \mathrm{sign}\left( \frac{1}{2}\theta^l \leq \boldsymbol{m}^l(t) \leq \left( L + \frac{1}{2} \right)\theta^l \right). \tag{8}$$

$$\frac{\partial \mathcal{L}}{\partial \lambda^l} = \sum_{t=1}^{T} \frac{\partial \mathcal{L}}{\partial \boldsymbol{m}^l(t)} \odot \boldsymbol{v}^l(t-1), \quad \frac{\partial \mathcal{L}}{\partial \omega_{ij}^l} = \frac{\partial \mathcal{L}}{\partial \boldsymbol{m}^l(i)} \odot \left( \boldsymbol{W}^l \boldsymbol{s}^{l-1}(j)\theta^{l-1} \right). \tag{9}$$

### 4.4 Hybrid Training based on the LM-HT Model

Although the traditional ANN-SNN Conversion frameworks have much lower computational overhead than STBP training algorithm, a serious performance degradation phenomenon often exists on the

Table 1: Ablation study for the LM-HT model on a subset of ImageNet-1k.

| Model | T-GIM | Arch. | Acc.(%) | SOPs(G) | E.(mJ) |
|-------|-------|-------|---------|---------|--------|
| L=1,T=4 | w/o | | 76.22 | 1.60 | 1.44 |
| L=2,T=2 | w/o | ResN-18 | 80.52 | 1.08 | 0.97 |
| L=2,T=2 | w/ | | 80.56 | 0.73 | 0.66 |
| L=1,T=4 | w/o | | 65.62 | 3.20 | 2.88 |
| L=2,T=2 | w/o | ResN-34 | 82.18 | 2.42 | 2.18 |
| L=2,T=2 | w/ | | 82.72 | 1.72 | 1.55 |

Table 2: Validation for the reparameterization procedure.

| Arch. | Acc.(%) | SOPs(G) | E.(mJ) |
|-------|---------|---------|--------|
| Before reparameterization (L=2, T=2) | | | |
| VGG-13 | 61.64 | 0.26 | 0.23 |
| ResN-18 | 64.44 | 0.52 | 0.46 |
| After reparameterization (L=1, T=4) | | | |
| VGG-13 | 61.66 | 0.26 | 0.23 |
| ResN-18 | 64.50 | 0.52 | 0.46 |

converted SNNs under low time latency [17]. To address this problem, previous researchers [39] considered adopting STBP training for a few epochs on the pre-trained ANN models to enhance the performance of the converted SNNs under fewer time-steps, which is called as hybrid training. In this work, we propose a brand-new hybrid training framework based on the LM-HT model.

We firstly choose QCFS function to train the quantized ANN models and then replace the QCFS function modules layer by layer with the LM-HT models under specific initialization ($\forall i, j \in [1, T], \omega_{ij}^l = \frac{1}{T}; \lambda^l = 1, \theta^l = \vartheta^l, \boldsymbol{v}^l(0) = \frac{\theta^l}{2}$), as shown in Fig.2(d). Combining with the conclusion pointed out by [2], one can note that the initialized LM-HT model and the QCFS function before substitution have an equivalence in terms of mathematical expectation, which has been described as the following theorem:

**Theorem 4.4.** When $\sum_{t=1}^{T} \boldsymbol{I}^l(t)/LT = \boldsymbol{W}^l \boldsymbol{r}_{IF}^{l-1}(T_q)$ and $\sum_{t=1}^{T} \boldsymbol{I}^l(t) \in [0, LT\theta^l]$, if $\forall i, j \in [1, T], \omega_{ij}^l = \frac{1}{T}$ and $\lambda^l = 1, \theta^l = \vartheta^l, \boldsymbol{v}^l(0) = \frac{\theta^l}{2}$, for $L, T, T_q$ with arbitrary values, we have:
$$\mathbb{E}\left( \frac{\sum_{t=1}^{T} \boldsymbol{s}^l(t)\theta^l}{LT} - \frac{\vartheta^l}{T_q} clip\left( \left\lfloor \frac{\boldsymbol{W}^l \boldsymbol{r}_{IF}^{l-1}(T_q)T_q}{\vartheta^l} + \frac{1}{2} \right\rfloor, 0, T_q \right) \right) = 0.$$

Theorem 4.4 indicates that regardless of whether the time-steps we choose during the STBP training phase is equal to the inference steps simulated in ANN-SNN Conversion, the average spike firing rate of the LM-HT models under the initial state of STBP training maintains a mathematical equivalence with that simulated by the QCFS function modules in the previous stage. Therefore, under this new training framework, we can adopt STBP algorithm to optimize the inference performance of SNN under any degree of time latency. The detailed pseudo-code has been provided in the Appendix.

### 4.5 Reparameterize the LM-HT model to vanilla LIF model

As discussed in Section 4.2, the mathematical essence of the LM-HT model is to simulate the spike firing situation of vanilla LIF neurons within each time window. Considering that the current neuromorphic hardware mainly supports single threshold models, we propose a reparameterization scheme that can transform the LM-HT model obtained during the training stage into a vanilla LIF model, which can further be deployed on hardware for inference.

As shown in Fig.3, for a $L$-level LM-HT model within $T$ steps, we expand it into a vanilla LIF model within $LT$ steps, where the membrane leakage factor between different time windows is set to $\lambda^l$. In addition, T-GIM will be extended from $\mathbb{R}^{T \times T}$ to $\mathbb{R}^{LT \times LT}$ and the parameters are averaged within each $L \times L$ sub-region, ensuring that the input current meets the precondition in Theorem 4.2. We also rectify the bias terms in synaptic layers, which involve addition operations at each time-step. By performing layer-by-layer reparameterization in the above manner, we will obtain a single threshold SNN model with theoretically lossless accuracy.

## 5 Experiments

To validate the effectiveness of our proposed STBP and hybrid training frameworks based on the LM-HT model, we consider multiple static and neuromorphic datasets with different data scale, including CIFAR-10(100) [23], ImageNet-200(1k) [6] and CIFAR10-DVS [26]. Consistent with the previous works, we also choose VGG [40] and ResNet [19] as the basic network architecture . We evaluate the computational overhead of SNNs based on the number of synaptic operations (SOPs) and the calculation standard for related energy consumption refers to [55]. In addition, as the information transmitted by our $L$-level LM-HT model within $T$ time-steps remains at the same level as that of the

Table 3: Comparison with previous state-of-the-art works.

| Dataset | Method | Type | Architecture | Time-steps | Accuracy(%) |
|---|---|---|---|---|---|
| CIFAR-10 | STBP-tdBN [54] | Direct Training | ResNet-19 | 4 | 92.92 |
| | Dspike [29] | Direct Training | ResNet-18 | 4 | 93.66 |
| | TET [8] | Direct Training | ResNet-19 | 4 | 94.44 |
| | SLTT [34] | Online Training | ResNet-18 | 6 | 94.44 |
| | GLIF [51] | Direct Training | ResNet-18 | 2, 4, 6 | 94.15, 94.67, 94.88 |
| | | | ResNet-19 | 2, 4, 6 | 94.44, 94.85, 95.03 |
| | **LM-HT (L=2)** | **Direct Training** | **ResNet-18** | **2** | **96.25** |
| | | | **ResNet-19** | **2** | **96.89** |
| CIFAR-100 | Dspike [29] | Direct Training | ResNet-18 | 4 | 73.35 |
| | TET [8] | Direct Training | ResNet-19 | 4 | 74.47 |
| | SLTT [34] | Online Training | ResNet-18 | 6 | 74.38 |
| | GLIF [51] | Direct Training | ResNet-18 | 2, 4, 6 | 74.60, 76.42, 77.28 |
| | | | ResNet-19 | 2, 4, 6 | 75.48, 77.05, 77.35 |
| | RMP-Loss [14] | Direct Training | ResNet-19 | 2, 4, 6 | 74.66, 78.28, 78.98 |
| | **LM-HT (L=2)** | **Direct Training** | **ResNet-18** | **2** | **79.33** |
| | | | **ResNet-19** | **2** | **81.76** |
| ImageNet-200 | DCT [13] | Hybrid Training | VGG-13 | 125 | 56.90 |
| | Online-LTL [48] | Hybrid Training | VGG-13 | 16 | 54.82 |
| | Offline-LTL [48] | | | 16 | 55.37 |
| | ASGL [44] | Direct Training | VGG-13 | 4, 8 | 56.57, 56.81 |
| | **LM-HT (L=2)** | **Direct Training** | **VGG-13** | **2, 4** | **61.09, 61.75** |
| | **LM-HT (L=4)** | | | **2** | **62.05** |
| ImageNet-1k | STBP-tdBN [54] | Direct Training | ResNet-34 | 6 | 63.72 |
| | TET [8] | Direct Training | ResNet-34 | 6 | 64.79 |
| | MBPN [15] | Direct Training | ResNet-34 | 4 | 64.71 |
| | RMP-Loss [14] | Direct Training | ResNet-34 | 4 | 65.17 |
| | SEW ResNet [11] | Direct Training | ResNet-34 | 4 | 67.04 |
| | GLIF [51] | Direct Training | ResNet-34 | 4 | 67.52 |
| | **LM-HT (L=2)** | **Direct Training** | **ResNet-34** | **2** | **70.90** |
| CIFAR10-DVS | STBP-tdBN [54] | Direct Training | ResNet-19 | 10 | 67.80 |
| | Dspike [29] | Direct Training | ResNet-18 | 10 | 75.40 |
| | MBPN [15] | Direct Training | ResNet-19 | 10 | 74.40 |
| | RMP-Loss [14] | Direct Training | ResNet-19 | 10 | 76.20 |
| | **LM-HT (L=2)** | **Direct Training** | **ResNet-18** | **2, 4** | **80.70, 81.00** |
| | **LM-HT (L=4)** | | | **2** | **81.90** |

vanilla LIF model within $LT$ time-steps, to make a fair evaluation, we will compare the performance of the $L$-level LM-HT model within $T$ steps with that of the previous works within $LT$ steps.

## 5.1 Ablation & Validation Studies for the LM-HT Model

As shown in Tab.1, we investigate the impact of threshold levels and T-GIM for our proposed model. One can note that vanilla IF neuron ($L = 1, T = 4$) is not well suited for deep networks (*e.g.* ResNet-34) and causes relatively high energy consumption, while the M-HT series models ($L = 2, T = 2$) can effectively overcome the performance degradation problem on deep networks. When we further utilize T-GIM to regulate global information on the time dimension, the learning ability of our model is enhanced and the computational overhead in synaptic layers is significantly reduced.

We also validate the feasibility about the reparameterization procedure mentioned above. As shown in Tab.2 and Fig.3, by copying and reparameterizing the parameters of synapses, T-GIM and LM-HT neurons layer by layer, we obtain a single threshold model that maintained almost the same performance and power consumption as the original LM-HT model. This convertible property enables the LM-HT model to be more flexibly deployed on neuromorphic hardware.

## 5.2 Comparison with Previous SoTA Works

We first investigate the competitiveness of our proposed model in the domain of STBP learning. As shown in Tab.3, our comparative works incorporate previous state-of-the-art (SoTA) methods in various sub-domains of STBP training, including batchnorm layer optimization [54, 15], improved surrogate gradients [29], learning function design [8, 14], energy-efficient training [13, 48, 34] and advanced neuron models [51, 44].

Table 4: The performance of hybrid training based on the LM-HT model for CIFAR-100 dataset.

| Method | Time-steps | VGG-16 | | ResNet-20 | |
|---|---|---|---|---|---|
| | | ANN Acc.(%) | SNN Acc.(%) | ANN Acc.(%) | SNN Acc.(%) |
| RMP [16] | 32, 64, 128 | 71.22 | - , - , 63.76 | 68.72 | 27.64, 46.91, 57.69 |
| SNM [43] | 32, 64, 128 | 74.13 | 71.80, 73.69, 73.95 | - | - |
| SRP [17] | 5, 6, 8 | 76.28 | 71.52, 74.31, 75.42 | 69.94 | 46.48, 53.96, 59.34 |
| QCFS ($T_q$=4) [2] | 2, 4, 8 | 76.11 | 63.33, 69.70, 74.12 | 63.90 | 38.04, 52.28, 61.77 |
| **LM-HT (L=2)** | **2** | - | **75.97 (+6.27)** | - | **63.55 (+11.27)** |
| | **4** | - | **76.49 (+2.37)** | - | **64.87 (+3.10)** |
| **LM-HT (L=4)** | **2** | - | **76.38 (+2.26)** | - | **63.43 (+1.66)** |
| QCFS ($T_q$=8) [2] | 2, 4, 8 | 77.31 | 64.85, 70.50, 74.63 | 69.56 | 19.76, 34.17, 55.50 |
| **LM-HT (L=2)** | **2** | - | **76.31 (+5.81)** | - | **67.08 (+32.91)** |
| | **4** | - | **76.79 (+2.16)** | - | **69.00 (+13.50)** |
| **LM-HT (L=4)** | **2** | - | **76.08 (+1.45)** | - | **67.21 (+11.71)** |

**CIFAR-10 & CIFAR-100.** For conventional static datasets, one can find that our solution demonstrates significant performance advantages. For ResNet-18 structure, we achieve the top-1 accuracies of 96.25% and 79.33% with merely 2 time-steps on CIFAR-10 and CIFAR-100 datasets, respectively. For ResNet-19 network with a larger parameter scale, our method fulfills the precisions of 96.89% and 81.76% within 2 time-steps, which at least outperforms other corresponding works with 2.04% and 3.48% under the same time latency. In addition, it is worth noting that our above results have even exceeded the performance of other works with more time-steps (*e.g.* 6 steps).

**ImageNet-200 & ImageNet-1k.** For large-scale datasets, we also confirm the superiority of the LM-HT model. For the two-level LM-HT model, we respectively reach the top-1 accuracies of 61.09% and 70.90% within 2 time-steps on ImageNet-200 and ImageNet-1k datasets, which is 4.52% higher than ASGL (4 steps) and 3.38% higher than GLIF (4 steps) under the same-level time overhead. For a larger training time-step, one can note that our method will also demonstrate a significant advantage. For example, the two-level LM-HT model reaches the precision of 61.75% with 4 time-steps, which has surpassed ASGL (8 steps) with 4.94%.

**CIFAR10-DVS.** We also evaluate the effectiveness of our approach on neuromorphic datasets. Compared to other previous methods, our proposed model can achieve better results on shallower networks with fewer time-steps. For instance, the two-level LM-HT model can achieve the accuracy of 80.70% after merely 2 time-steps.

### 5.3 Performance Analysis of Hybrid Training

In our hybrid training framework, we first choose [2] as the backbone for our ANN-SNN Conversion stage. Subsequently, we replace the QCFS function layer by layer with the initialized LM-HT model and conduct STBP training for merely 30 epochs. Furthermore, we also consider other advanced conversion methods [16, 43] and multi-stage error correction method [17] as our comparative works.

As shown in Tab.4, after conducting the STBP fine-tuning optimization with relatively low computational overhead, we note that the performance of the converted SNNs under different quantization levels has been significantly improved and surpass other previous methods, especially under low time latency. For instance, compared to the ResNet-20 network after eight-level quantization (*i.e.* $T_q$=8), the two-level LM-HT model has achieved a performance improvement of 32.91% and 13.50% with 2 and 4 time-steps, respectively.

## 6 Conclusions

In this paper, we first investigate the mathematical equivalence among the multi-threshold model, vanilla spiking model and quantized ANNs, then propose an advanced STBP training method based on the LM-HT model, which has been proven to cover the representation range of vanilla STBP and quantized ANNs training frameworks, thereby promoting SNNs to achieve superior performance at the same level as quantized ANNs. Furthermore, the LM-HT model can achieve lossless transformation towards single threshold models or quantized ANNs under specific parameter configuration. Numerous experimental results have verified the effectiveness of our method. We believe that our work will further promote in-depth research on advanced spiking neural model.

## Acknowledgements

This work was supported by STI 2030-Major Projects 2021ZD0200300, the National Natural Science Foundation of China under Grant No. 62176003 and No. 62088102, and by Beijing Nova Program under Grant No. 20230484362.

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

# A  Appendix

## A.1  Proof of Theorem

### A.1.1  Proof of Lemma 4.1 & Theorem 4.2

Before the proof of Theorem 4.2, we first need to introduce Lemma A.1:

**Lemma A.1.** *Assume a continuous $T$-step input current $\boldsymbol{I}^l(1), ..., \boldsymbol{I}^l(T)$, for a LM-HT model with $L$-level threshold, when $\forall t \in [1, T], \boldsymbol{I}^l(t) \in [0, L\theta^l)$ and $\boldsymbol{v}^l(0) \in [0, \theta^l), \lambda^l = 1$, we will have $\boldsymbol{v}^l(T) \in [0, \theta^l)$.*

*Proof.* $\forall t \in [0, T)$, if $\boldsymbol{v}^l(t) \in [0, \theta^l)$, as $\boldsymbol{m}^l(t+1) = \boldsymbol{v}^l(t) + \boldsymbol{I}^l(t)$, we have $\boldsymbol{m}^l(t+1) \in [0, (L+1)\theta^l)$. Therefore, after the firing process $\boldsymbol{v}^l(t+1) = \boldsymbol{m}^l(t+1) - \boldsymbol{s}^l(t)\theta^l$, one can note that $\boldsymbol{v}^l(t+1) \in [0, \theta^l)$. According to the idea of mathematical induction, if we directly set $\boldsymbol{v}^l(0) \in [0, \theta^l)$, we can have $\boldsymbol{v}^l(T) \in [0, \theta^l)$. $\square$

**Theorem 4.2.** *When $\lambda^l = 1, \boldsymbol{v}^l(0) \in [0, \theta^l)$, for a M-HT model with $L$-level threshold, after $T$ time-steps, we will derive the following conclusions:*
*(i) If we further assume $\forall t \in [1, T], \boldsymbol{I}^l(t) \in [0, L\theta^l)$, we will have: $\forall t \in [1, T], \boldsymbol{s}^l(t) = \sum_{j=L(t-1)+1}^{Lt} \boldsymbol{s}_{IF}^l(j), \boldsymbol{v}^l(t) = \boldsymbol{v}_{IF}^l(Lt), \sum_{t=1}^{T} \boldsymbol{s}^l(t) = \sum_{j=1}^{LT} \boldsymbol{s}_{IF}^l(j)$.*
*(ii) If we further assume $\boldsymbol{I}^l(1) = ... = \boldsymbol{I}^l(T)$, we will have: $\sum_{t=1}^{T} \boldsymbol{s}^l(t) = \mathrm{clip}\left(\left\lfloor \frac{\boldsymbol{v}^l(0) + \sum_{t=1}^{T} \boldsymbol{I}^l(t)}{\theta^l} \right\rfloor, 0, LT\right)$.*
*Here the IF model has uniform input currents $\boldsymbol{I}^l(1)/L, ..., \boldsymbol{I}^l(T)/L$ respectively within every $L$ steps and satisfies $\boldsymbol{v}_{IF}^l(0) = \boldsymbol{v}^l(0)$.*

*Proof.* (i) If we consider the pre-condition in Theorem 4.2 and combine Eq.(1) with Eq.(2), $\forall t \in [1, LT]$, we will have:

$$\boldsymbol{v}_{IF}^l(t) - \boldsymbol{v}_{IF}^l(t-1) = \boldsymbol{I}^l\left(\left\lceil \frac{t}{L} \right\rceil\right)/L - \boldsymbol{s}_{IF}^l(t)\theta^l. \tag{S1}$$

Similarly, if we set $\lambda^l = 1$ and incorporate Eq.(5), $\forall t \in [1, T]$, we will have:

$$\boldsymbol{v}^l(t) - \boldsymbol{v}^l(t-1) = \boldsymbol{I}^l(t) - \boldsymbol{s}^l(t)\theta^l. \tag{S2}$$

Then we accumulate Eq.(S1) along the time dimension and obtain the following equation:

$$\boldsymbol{v}_{IF}^l(Lt) - \boldsymbol{v}_{IF}^l(L(t-1)) = \boldsymbol{I}^l(t) - \sum_{j=L(t-1)+1}^{Lt} \boldsymbol{s}_{IF}^l(j)\theta^l. \tag{S3}$$

As $\boldsymbol{I}^l(t) \in [0, L\theta^l)$, according to Lemma A.1, when $\boldsymbol{v}^l(t-1) = \boldsymbol{v}_{IF}^l(L(t-1)) \wedge \boldsymbol{v}^l(t-1) \in [0, \theta^l)$, we will have $\boldsymbol{v}^l(t) \in [0, \theta^l)$ and $\boldsymbol{v}_{IF}^l(Lt) \in [0, \theta^l)$. Considering $\boldsymbol{s}^l(t), \sum_{j=L(t-1)+1}^{Lt} \boldsymbol{s}_{IF}^l(j) \in \mathbb{N}$, if $\boldsymbol{s}^l(t) \neq \sum_{j=L(t-1)+1}^{Lt} \boldsymbol{s}_{IF}^l(j)$, one can note that $|\sum_{j=L(t-1)+1}^{Lt} \boldsymbol{s}_{IF}^l(j)\theta^l - \boldsymbol{s}^l(t)\theta^l| = |(\boldsymbol{v}^l(t) - \boldsymbol{v}^l(t-1)) - (\boldsymbol{v}_{IF}^l(Lt) - \boldsymbol{v}_{IF}^l(L(t-1)))| = |\boldsymbol{v}^l(t) - \boldsymbol{v}_{IF}^l(Lt)| \geq \theta^l$, which will violate the conclusion in Lemma A.1. Therefore, we can finally deduce that $\boldsymbol{s}^l(t) = \sum_{j=L(t-1)+1}^{Lt} \boldsymbol{s}_{IF}^l(j)$. Then we can further have $\boldsymbol{v}^l(t) = \boldsymbol{v}_{IF}^l(Lt)$ and $\sum_{t=1}^{T} \boldsymbol{s}^l(t) = \sum_{j=1}^{LT} \boldsymbol{s}_{IF}^l(j)$.

(ii) If we accumulate Eq.(S2) along the time dimension and divide $\theta^l$ on both sides, we will have the following equation:

$$\frac{\boldsymbol{v}^l(T) - \boldsymbol{v}^l(0)}{\theta^l} = \frac{\sum_{t=1}^{T} \boldsymbol{I}^l(t)}{\theta^l} - \sum_{t=1}^{T} \boldsymbol{s}^l(t). \tag{S4}$$

If $\boldsymbol{I}^l(1) < 0$ or $\boldsymbol{I}^l(1) \geq L\theta^l$, it is obvious that we will have $\sum_{t=1}^{T} \boldsymbol{s}^l(t) = \mathrm{clip}\left(\left\lfloor \frac{\boldsymbol{v}^l(0) + \sum_{t=1}^{T} \boldsymbol{I}^l(t)}{\theta^l} \right\rfloor, 0, LT\right) = 0$ or $\sum_{t=1}^{T} \boldsymbol{s}^l(t) = \mathrm{clip}\left(\left\lfloor \frac{\boldsymbol{v}^l(0) + \sum_{t=1}^{T} \boldsymbol{I}^l(t)}{\theta^l} \right\rfloor, 0, LT\right) = L.$

If $\boldsymbol{I}^l(1) \in [0, L\theta^l)$, according to Lemma A.1, we will have $\boldsymbol{v}^l(T) \in [0, \theta^l)$. As $\sum_{t=1}^{T} \boldsymbol{s}^l(t) \in \mathbb{N}$, based on Eq.(S4), we can finally deduce that $\sum_{t=1}^{T} \boldsymbol{s}^l(t) = \frac{\boldsymbol{v}^l(0)+\sum_{t=1}^{T} \boldsymbol{I}^l(t)}{\theta^l} - \frac{\boldsymbol{v}^l(T)}{\theta^l} = \left\lfloor \frac{\boldsymbol{v}^l(0)+\sum_{t=1}^{T} \boldsymbol{I}^l(t)}{\theta^l} \right\rfloor = \text{clip}\left(\left\lfloor \frac{\boldsymbol{v}^l(0)+\sum_{t=1}^{T} \boldsymbol{I}^l(t)}{\theta^l} \right\rfloor, 0, LT\right)$.

One can note that Lemma 4.1 is actually a special case of Theorem 4.2 under the condition of $T = 1$, therefore Lemma 4.1 is also proven. $\qquad\square$

### A.1.2 Proof of Corollary 4.3

**Corollary 4.3.** *If* $\lambda^l = 1, \boldsymbol{v}^l(0) = 0$ *and* $\boldsymbol{I}^l(1) = ... = \boldsymbol{I}^l(T)$, *for a M-HT model with L-level threshold,* $\boldsymbol{s}^l(1) = ... = \boldsymbol{s}^l(T)$ *is only satisfied when* $\boldsymbol{I}^l(1) \in [k\theta^l, k\theta^l + \theta^l/T), \forall k = 0, ..., L-1$ *or* $\boldsymbol{I}^l(1) \in (-\infty, 0) \cup [L\theta^l, +\infty)$.

*Proof.* If $\boldsymbol{I}^l(1) < 0$ or $\boldsymbol{I}^l(1) \geq L\theta^l$, it is obvious that we will have $\boldsymbol{s}^l(1) = ... = \boldsymbol{s}^l(T) = 0$ or $\boldsymbol{s}^l(1) = ... = \boldsymbol{s}^l(T) = L$. Otherwise, based on the conclusion $\sum_{t=1}^{T} \boldsymbol{s}^l(t) = \text{clip}\left(\left\lfloor \frac{\boldsymbol{v}^l(0)+\sum_{t=1}^{T} \boldsymbol{I}^l(t)}{\theta^l} \right\rfloor, 0, LT\right)$ in Theorem 4.2(ii), when $\boldsymbol{I}^l(1) \in [k\theta^l, k\theta^l + \theta^l/T), \forall k = 0, ..., L-1$, we will have $\sum_{t=1}^{T} \boldsymbol{s}^l(t) = kT, \forall k = 0, ..., L-1$. Note that $\forall T' \in [1, T]$, we can further have $\sum_{t=1}^{T'} \boldsymbol{s}^l(t) = kT', \forall k = 0, ..., L-1$. Therefore, it can be concluded that $\boldsymbol{s}^l(1) = ... = \boldsymbol{s}^l(T) = k$. Instead, if $\boldsymbol{I}^l(1) \in [0, L\theta^l) \wedge \boldsymbol{I}^l(1) \notin [k\theta^l, k\theta^l + \theta^l/T), \forall k = 0, ..., L-1$, we will have $\sum_{t=1}^{T} \boldsymbol{s}^l(t) \neq kT, \forall k = 0, ..., L-1$. Therefore, $\boldsymbol{s}^l(1) = ... = \boldsymbol{s}^l(T)$ does not hold true.

$\qquad\square$

### A.1.3 Proof of Theorem 4.4

**Theorem 4.4.** *When* $\sum_{t=1}^{T} \boldsymbol{I}^l(t)/LT = \boldsymbol{W}^l \boldsymbol{r}_{IF}^{l-1}(T_q)$ *and* $\sum_{t=1}^{T} \boldsymbol{I}^l(t) \in [0, LT\theta^l]$, *if* $\forall i, j \in [1, T], \omega_{ij}^l = \frac{1}{T}$ *and* $\lambda^l = 1, \theta^l = \vartheta^l, \boldsymbol{v}^l(0) = \frac{\theta^l}{2}$, *for* $L, T, T_q$ *with arbitrary values, we have:*
$$\mathbb{E}\left(\frac{\sum_{t=1}^{T} \boldsymbol{s}^l(t)\theta^l}{LT} - \frac{\vartheta^l}{T_q}\text{clip}\left(\left\lfloor \frac{\boldsymbol{W}^l \boldsymbol{r}_{IF}^{l-1}(T_q)T_q}{\vartheta^l} + \frac{1}{2} \right\rfloor, 0, T_q\right)\right) = 0.$$

*Proof.* If $\forall i, j \in [1, T], \omega_{ij}^l = \frac{1}{T}, \lambda^l = 1, \theta^l = \vartheta^l$ and $\boldsymbol{v}^l(0) = \frac{\theta^l}{2}$, combining with the conclusion mentioned in Theorem 4.2(ii), we will have $\frac{\sum_{t=1}^{T} \boldsymbol{s}^l(t)\theta^l}{LT} = \frac{\theta^l}{LT}\text{clip}\left(\left\lfloor \frac{\sum_{t=1}^{T} \boldsymbol{I}^l(t)}{\theta^l} + \frac{1}{2} \right\rfloor, 0, LT\right)$. According to the conclusion pointed out in [2], we have known that $\mathbb{E}\left(\frac{\theta^l}{LT}\text{clip}\left(\left\lfloor \frac{\boldsymbol{x}^l LT}{\theta^l} + \frac{1}{2} \right\rfloor, 0, LT\right) - \frac{\vartheta^l}{T_q}\text{clip}\left(\left\lfloor \frac{\boldsymbol{x}^l T_q}{\vartheta^l} + \frac{1}{2} \right\rfloor, 0, T_q\right)\right) = 0$, here $\boldsymbol{x}^l \in [0, \theta^l]$. Therefore, we directly set $\boldsymbol{x}^l = \sum_{t=1}^{T} \boldsymbol{I}^l(t)/LT = \boldsymbol{W}^l \boldsymbol{r}_{IF}^{l-1}(T_q)$ and then we will draw the final conclusion.

$\qquad\square$

### A.1.4 Computational Equivalence about the Reparameterization Process

**Theorem A.2.** $\forall t, i \in [1, T], \forall j \in [L(t-1)+1, Lt], \forall k \in [L(i-1)+1, Li]$, *when* $\boldsymbol{s}^{l-1}(t) = \sum_{j=L(t-1)+1}^{Lt} \boldsymbol{s}_{IF}^{l-1}(j)$, *if* $\hat{b}_j^l = b_t^l/L, \hat{\omega}_{jk}^l = \omega_{ti}^l/L$, *we will have* $\boldsymbol{I}^l(t) = \sum_{j=L(t-1)+1}^{Lt} \boldsymbol{I}_{IF}^l(j)$. *Here* $\hat{b}^l, \hat{\omega}^l$ *denote the rectified bias term and T-GIM layer after the reparameterization process.*

*Proof.* Firstly, it is obvious that $\boldsymbol{I}^l(t), \boldsymbol{I}_{IF}^l(j)$ can be rewritten as $\boldsymbol{I}^l(t) = \sum_{i=1}^{T} \omega_{ti}^l(\boldsymbol{W}^l \boldsymbol{s}^{l-1}(i) + b_i^l)$ and $\boldsymbol{I}_{IF}^l(j) = \sum_{i=1}^{LT} \hat{\omega}_{ji}^l(\boldsymbol{W}^l \boldsymbol{s}_{IF}^{l-1}(i) + \hat{b}_i^l)$. Considering the precondition $\hat{b}_j^l = b_t^l/L, \hat{\omega}_{jk}^l = \omega_{ti}^l/L$,

we will have:

$$\boldsymbol{I}^l_{IF}(j) = \sum_{i=1}^{T} \sum_{k=L(i-1)+1}^{Li} \hat{\omega}^l_{jk}(\boldsymbol{W}^l \boldsymbol{s}^{l-1}_{IF}(k) + \hat{b}^l_k)$$

$$= \sum_{i=1}^{T} \frac{\omega^l_{ti}}{L} \sum_{k=L(i-1)+1}^{Li} (\boldsymbol{W}^l \boldsymbol{s}^{l-1}_{IF}(k) + \frac{b^l_i}{L}). \tag{S5}$$

Then we can further have:

$$\sum_{j=L(t-1)+1}^{Lt} \boldsymbol{I}^l_{IF}(j) = \sum_{j=L(t-1)+1}^{Lt} \sum_{i=1}^{T} \frac{\omega^l_{ti}}{L} \sum_{k=L(i-1)+1}^{Li} (\boldsymbol{W}^l \boldsymbol{s}^{l-1}_{IF}(k) + \frac{b^l_i}{L})$$

$$= \sum_{i=1}^{T} \frac{\omega^l_{ti}}{L} \sum_{j=L(t-1)+1}^{Lt} (\boldsymbol{W}^l \sum_{k=L(i-1)+1}^{Li} \boldsymbol{s}^{l-1}_{IF}(k) + b^l_i)$$

$$= \sum_{i=1}^{T} \frac{\omega^l_{ti}}{L} \sum_{j=L(t-1)+1}^{Lt} (\boldsymbol{W}^l \boldsymbol{s}^{l-1}(i) + b^l_i)$$

$$= \sum_{i=1}^{T} \omega^l_{ti}(\boldsymbol{W}^l \boldsymbol{s}^{l-1}(i) + b^l_i)$$

$$= \boldsymbol{I}^l(t). \tag{S6}$$

Due to the fact that the calculation process of the spike sequences passing through Conv & BN and T-GIM layers can be abstractly described by Theorem A.2, we can conclude that the sum of the input currents within the corresponding time windows before and after reparameterization remains unchanged. The spike sequences obtained by passing the input currents through the spiking neuron layer will also satisfy the precondition of Theorem A.2 ($\boldsymbol{s}^l(t) = \sum_{j=L(t-1)+1}^{Lt} \boldsymbol{s}^l_{IF}(j)$). Therefore, we can prove the computational equivalence before and after the reparameterization process. $\square$

## A.2 Comparison with Other Advanced Network Backbones

As shown in Tab.S1, we have made comparison with related advanced works [20, 55, 49, 38] on CIFAR datasets. One can find that our LM-HT model has superior scalability and can demonstrate its effectiveness on multiple different backbones. For example, for CIFAR-100 dataset, compared to GAC-SNN [38], we achieve an accuracy improvement of 1.35% on MS-ResNet-18. For Transformer-4-384 architecture, our method also outperforms Spikformer [55] and Spike-driven Transformer [49] in terms of performance.

Table S1: Comparison with previous methods based on advanced backbones and attention mechanism.

| Dataset | Method | Architecture | Time-steps | Accuracy(%) |
|---------|--------|--------------|------------|-------------|
| CIFAR-10 | MS-ResNet [20] | MS-ResNet-18 | 6 | 94.92 |
| | GAC-SNN [38] | MS-ResNet-18 | 4 | 96.24 |
| | Spikformer [55] | Transformer-4-384 | 4 | 95.51 |
| | Spike-driven Transformer [49] | Transformer-4-384 | 4 | 95.6 |
| | **Ours** | **MS-ResNet-18** | **4** | **96.38** |
| | | **Transformer-4-384** | **4** | **95.82** |
| CIFAR-100 | MS-ResNet [20] | MS-ResNet-18 | 6 | 76.41 |
| | GAC-SNN [38] | MS-ResNet-18 | 4 | 79.83 |
| | Spikformer [55] | Transformer-4-384 | 4 | 78.21 |
| | Spike-driven Transformer [49] | Transformer-4-384 | 4 | 78.4 |
| | **Ours** | **MS-ResNet-18** | **4** | **81.18** |
| | | **Transformer-4-384** | **4** | **79.03** |

## A.3 Experimental Configuration

For static datasets, we attempt to suppress the possible overfitting phenomenon by utilizing data augmentation techniques including AutoAugment [4] and Cutout [9]. For CIFAR10-DVS dataset, we resize each image to $48 \times 48$ pixels and split it into 10 frames. For ImageNet-1k dataset, we further consider MS-ResNet architecture [20] and Mixup technique [53] to strengthen the generalization ability of our network. We respectively try to use SGD [1] and AdamW [30] as our optimizers. The corresponding initial learning rate and weight decay are set to $0.025, 5 \times 10^{-4}$ for SGD on CIFAR-10(100), $0.0125, 5 \times 10^{-4}$ for SGD on ImageNet-200 and $0.02, 0.01$ for AdamW on CIFAR10-DVS. For ImageNet-1k dataset, we use SGD as our optimizer and set the corresponding weight decay as $0$. Furthermore, in the hybrid training framework, our initial learning rate and weight decay are both set to $5 \times 10^{-4}$. The QCFS pretrained models are selected from related open-source checkpoints and self-implementation. For all experimental cases, we choose the Cosine Annealing scheduler [31] to dynamically regulate the learning rate. Our experiments are implemented on NVIDIA RTX A5000 and 4090.

## A.4 The Pseudo-Code of Hybrid Training Algorithm

---

**Algorithm 1** Hybrid training framework based on the LM-HT model.

---

**Require:** Pretrained QCFS ANN model $f_{\text{ANN}}(\boldsymbol{W}, T_q, \vartheta)$ with $L_N$ layers; Dataset $D$; Number of time-steps choosed for STBP training $T$.
**Ensure:** SNN model $f_{\text{SNN}}(\boldsymbol{W}, \boldsymbol{\Omega}, L, \lambda, \theta)$.
 1: # Convert ANN to SNN
 2: **for** $l = 1$ to $L_N$ **do**
 3:     $f_{\text{SNN}}.\boldsymbol{W}^l = f_{\text{ANN}}.\boldsymbol{W}^l$
 4:     $f_{\text{SNN}}.\theta^l = f_{\text{ANN}}.\vartheta^l$
 5:     $f_{\text{SNN}}.\boldsymbol{\Omega}^l = \frac{1}{T}$
 6:     $f_{\text{SNN}}.\lambda^l = 1$
 7:     $f_{\text{SNN}}.\boldsymbol{v}^l(0) = f_{\text{SNN}}.\theta^l/2$
 8: **end for**
 9: # STBP training based on the LM-H model
10: # Set $f_{\text{SNN}}.\boldsymbol{\Omega}^l, f_{\text{SNN}}.\lambda^l$ as learnable parameters and $f_{\text{SNN}}.\theta^l$ as scalars
11: **for** (**Image**,**Label**) in $D$ **do**
12:     **for** $l = 1$ to $L_N$ **do**
13:         **if** Is the first layer **then**
14:             **for** $t = 1$ to $T$ **do**
15:                 $\boldsymbol{I}^l(t) = \boldsymbol{I}^l(t) \times L$
16:             **end for**
17:         **end if**
18:         LM-HT model performs forward propagation based on Eqs.(5)-(6) and Eq.(7)
19:         LM-HT model performs back-propagation based on Eqs.(8)-(9)
20:     **end for**
21: **end for**
22: **return** $f_{\text{SNN}}(\boldsymbol{W}, \boldsymbol{\Omega}, L, \lambda, \theta)$

---

