# OpenReview forum: "LM-HT SNN: Enhancing the Performance of SNN to ANN Counterpart through Learnable Multi-hierarchical Threshold Model"
_NeurIPS.cc/2024/Conference — NeurIPS 2024 poster_

### Official Review · Reviewer_59n9 · 2024-06-18

**Soundness:** 3
**Presentation:** 3
**Contribution:** 3
**Rating:** 7
**Confidence:** 5

**Summary:**

This paper introduces a novel learning model for SNNs that dynamically adjusts input current and membrane potential leakage, enhancing SNN performance to match ANNs. The proposed LM-HT model can be seamlessly integrated with ANN-SNN Conversion frameworks, effectively improving the performance of converted SNNs under low time latency. Extensive experiments demonstrate the model's superior performance on various datasets, achieving state-of-the-art results.

**Strengths:**

The paper's strengths include the introduction of the LM-HT model, which significantly enhances SNN performance through dynamic regulation of input current and membrane potential leakage. It also presents a seamless integration with ANN-SNN Conversion frameworks, leading to improved performance under low time latency. Additionally, the extensive experimental validation demonstrates the model's state-of-the-art performance across multiple datasets.

**Weaknesses:**

1. Checklist should be placed after appendix.
2. The authors can also compare with paper [1][2] which obtains really good results by MS-ResNet-34 backbone on large imageNet datasets.
3. The paper might not address the scalability of the LM-HT model to different SNN backbone. The authors can conduct experiments on more backbones (MS-ResNet [3] and Spikformer [4][5]) to demonstrate the effectiveness of the proposed LM-HT SNN.
4. While the LM-HT model can be reparameterized to a single-threshold model for inference, the paper may not fully explore the implications of this transformation for deployment on neuromorphic hardware.




[1] Yao M, Zhao G, Zhang H, et al. Attention spiking neural networks[J]. IEEE transactions on pattern analysis and machine intelligence, 2023.

[2] Qiu X, Zhu R J, Chou Y, et al. Gated attention coding for training high-performance and efficient spiking neural networks[C]. Proceedings of the AAAI Conference on Artificial Intelligence. 2024, 38(1): 601-610.

[3]Hu Y, Deng L, Wu Y, et al. Advancing Spiking Neural Networks Toward Deep Residual Learning[J]. IEEE Transactions on Neural Networks and Learning Systems, 2024.

[4] Yao M, Hu J, Zhou Z, et al. Spike-driven transformer[J]. Advances in Neural Information Processing Systems, 2024, 36.

[5] Zhou Z, Zhu Y, He C, et al. Spikformer: When Spiking Neural Network Meets Transformer[C]//The Eleventh International Conference on Learning Representations. 2022.

**Questions:**

See  Weaknesses.

**Limitations:**

The author has discussed the limitations.

---

> ### Author Rebuttal · Authors · 2024-08-05
>
> ## To Reviewer 59n9
> Thanks for your valuable and constructive feedback. We are encouraged that you found our paper "significantly enhances SNN performance", "presents a seamless integration with ANN-SNN Conversion frameworks" and "has extensive experimental validation". We would like to address your concerns and answer your questions in the following.
>
> > 1. Checklist should be placed after appendix.
>
> Thanks for pointing it out. We will make relevant revision in the final version.
>
> > 2. The authors can conduct experiments on more backbones (MS-ResNet [3] and Spikformer [4, 5]) to demonstrate the effectiveness of the proposed LM-HT SNN. The authors can also compare with papers [1, 2] which obtain really good results by MS-ResNet-34 backbone on large imageNet datasets.
>
> Thanks for your suggestions! As shown in Table R1-R2, we have made comparison with related advanced works [1, 2, 3, 4, 5] on CIFAR and ImageNet datasets. One can find that our LM-HT model has superior scalability and can demonstrate its effectiveness on multiple different backbones.
>
> For example, for CIFAR-100 and ImageNet-1k datasets, compared to GAC-SNN [2], we achieve accuracy improvements of 1.35% and 1.13% on MS-ResNet-18/34, respectively. For Transformer-4-384 architecture, our method also outperforms Spikformer [5] and Spike-driven Transformer [4] in terms of performance.
>
> **Table R1: Comparison with MS-ResNet series works [1, 2, 3].**
>
> | Dataset | Arch. | time-steps | MS-ResNet [3] | Att-MS-ResNet [1] | GAC-SNN [2] | **Ours** |
> | --- | --- | --- | --- | --- | --- | --- |
> | CIFAR-10 | MS-ResNet-18 | 4 | 94.92 (6 steps) | - | 96.24 | **96.38** |
> | CIFAR-100 | MS-ResNet-18 | 4 | 76.41 (6 steps) | - | 79.83 | **81.18** |
> | ImageNet-1k | MS-ResNet-34 | 4 | 69.43 (6 steps) | 69.35 (6 steps) | 69.77 | **70.90** |
>
> **Table R2: Comparison with Spikformer series works [4, 5].**
>
> | Dataset | Arch. | time-steps | Spikformer [5] | Spike-driven Transformer [4] | **Ours** |
> | --- | --- | --- | --- | --- | --- |
> | CIFAR-10 | Transformer-4-384 | 4 | 95.51 | 95.6 | **95.82** |
> | CIFAR-100 | Transformer-4-384 | 4 | 78.21 | 78.4 | **79.03** |
>
> > 3. While the LM-HT model can be reparameterized to a single-threshold model for inference, the paper may not fully explore the implications of this transformation for deployment on neuromorphic hardware.
>
> Thank for the question. As shown in Fig.3, the main differences before and after reparameterization are the adjustments to the bias terms of Conv&BN layers, the shape of T-GIM layers, and the membrane leakage parameters，which aims to keep the sum of the input currents within each time window during the inference phase consistent with the expected input currents at the corresponding time-step during the training phase, thus further ensuring that the sum of spike sequences emitted by each layer within each time window is consistent with the multi-bit spikes emitted at the corresponding time-step during the training phase.
>
> Besides these, there have been no changes in other parts of the network. Therefore, the reparameterized LM-HT SNN has consistent network architecture and firing mechanism with the SNN based on the vanilla LIF model, and we believe it can be effectively deployed on neuromorphic hardware.
> The experimental results in Table 2 also confirm that after being reparameterized and converted to a vanilla single-threshold model, the LM-HT model can maintain consistent performance and SOPs with the multi-threshold model before the reparameterization procedure.
>
>
> [1] Yao M, Zhao G, Zhang H, et al. Attention spiking neural networks[J]. IEEE transactions on pattern analysis and machine intelligence, 2023.
>
> [2] Qiu X, Zhu R J, Chou Y, et al. Gated attention coding for training high-performance and efficient spiking neural networks[C]. Proceedings of the AAAI Conference on Artificial Intelligence. 2024.
>
> [3] Hu Y, Deng L, Wu Y, et al. Advancing Spiking Neural Networks Toward Deep Residual Learning[J]. IEEE Transactions on Neural Networks and Learning Systems, 2024.
>
> [4] Yao M, Hu J, Zhou Z, et al. Spike-driven transformer[J]. Advances in Neural Information Processing Systems, 2024.
>
> [5] Zhou Z, Zhu Y, He C, et al. Spikformer: When Spiking Neural Network Meets Transformer[C]. The Eleventh International Conference on Learning Representations, 2022.

---

> > ### Comment · Reviewer_59n9 · 2024-08-08
> >
> > Thank you for your reply. I think this is a nice bit of discussion and could be added to the manuscript. In light of the additional discussion, I'd like to raise my score to a 7. This is an interesting piece of work and would be a nice addition to NeurIPS.

---

### Official Review · Reviewer_wxgA · 2024-07-03

**Soundness:** 4
**Presentation:** 3
**Contribution:** 3
**Rating:** 7
**Confidence:** 4

**Summary:**

Traditional SNNs adopt binary spike communications, resulting in relatively poor performance compared to their ANN counterparts. In this work, the authors propose an advanced Multi-hierarchical Threshold (LM-HT) Model. In my view, LM-HT models introduce several low-bit representation precisions to model the spiking number during short time periods. This perspective not only brings a nice mathematical relationship with quantized ANN training but also enhances SNN performance on several image classification tasks and certain types of neuromorphic datasets.

**Strengths:**

The LM-HT offers a nice perspective on how to further improve SNN performance with solid theoretical analysis. The overall idea makes sense to me, and the writing is generally good.

**Weaknesses:**

- It is less significant to only report the best accuracy. The authors are encouraged to provide mean accuracy and standard deviation over at least three different seeds for statistical significance.
- The use of multi-precision spikes in SNNs has been reported in several previous studies. To cater to a broader background of readers, the authors should better clarify their motivations, either from the biological rationale or performance perspective.

Minor issues:
- The authors should correct the name of DVSCIFAR-10 and several typos, such as “with larger parameter scale” to “with a larger parameter scale.”

**Questions:**

The authors lack sufficient disucssions for its limiations. The proposed methods seem to be hard to extend to a broader range of sequential learning tasks.

**Limitations:**

-	In Table 3, the types of learning methods should be well explained in the caption. What is the meaning of efficient training?

- It is not straightforward to understand the time-steps in CIFAR10-DVS, as it is normally assumed to classify a gesture sequence. Given such high performance with only two time steps, have the authors tested the results with a single time step?

---

> ### Author Rebuttal · Authors · 2024-08-05
>
> ## To Reviewer wxgA
> Thanks for your insightful and valuable feedback. We are glad that you found our work "offers a nice perspective", "has solid theoretical analysis" and "the writing is generally good". We would like to address your concerns and questions in the following.
>
> > 1. The authors are encouraged to provide mean accuracy and standard deviation over at least three different seeds for statistical significance.
>
> Thanks for the question. Due to time constraints, we conducted 5 rounds of validation using different random seeds on CIFAR-10/ResNet-18 and obtained an statistical accuracy result of $96.20$% ± $0.10$%, which can verify the superiority of our method.
>
> > 2. The use of multi-precision spikes in SNNs has been reported in several previous studies. To cater to a broader background of readers, the authors should better clarify their motivations.
>
> Thanks for your suggestion. We think that the core motivation and contribution of this paper lies in revealing the relationship between multi-threshold models, vanilla spiking neurons and quantized ANNs, thus establishing a bridge between ANN-SNN Conversion and STBP training, which are two core training algorithms in the SNN community, while previous works focused more on using multi-precision spikes from the perspective of enhancing the richness of information transmission or reducing inference errors in ANN-SNN Conversion learning. We think that this point might be overlooked in previous works based on multi-precision spikes.
>
> > 3. The authors should correct the name of DVSCIFAR-10 and several typos.
>
> Thanks for pointing it out. We will revise them in the final submitted version.
>
> > 4. The proposed methods seem to be hard to extend to a broader range of sequential learning tasks.
>
> Thanks for the question. As shown in Eqs.(8)-(9), we designed the the back-propagation calculation chain of the LM-HT model in a form similar to quantized ANNs (detach the term $\frac{\partial m^l(t)}{\partial m^l(t-1)}$ and establish the correlation calculation of gradients at different time-steps solely through the T-GIM layer), which means that the LM-HT model is more inclined to learn global sequential information on the time dimension.
>
> We think that if there is a need to learn towards long-term or local temporal information, one possible solution is to consider adjusting the back-propagation mode of the LM-HT model to the computational mode of vanilla LIF model in STBP training (consider the temporal back-propagation term $\frac{\partial m^l(t)}{\partial m^l(t-1)}$), as shown in Eq.(3).
>
> > 5. In Table 3, the types of learning methods should be well explained in the caption. What is the meaning of efficient training?
>
> Thanks for pointing it out. The term "efficient training" here actually refers to online training, which means that the backward gradient will be updated immediately after the forward propagation is completed at a specific time-step, thereby keeping the GPU memory overhead at a constant level. we will make relevant modification in the final version.
>
> > 6. It is not straightforward to understand the time-steps in CIFAR10-DVS, as it is normally assumed to classify a gesture sequence. Given such high performance with only two time steps, have the authors tested the results with a single time step?
>
> Thanks for the question. As we mentioned in Appendix A.2, we will still divide CIFAR10-DVS into 10 frames, while the T-GIM layer located before the first LM-HT neural layer will compress these 10 frames into the required time-steps $T'$ ($\Omega^1\in \mathbb{R}^{T'\times T}$). The experimental results in Table 3 indicate that good performance can be achieved on CIFAR10-DVS with just such a simple treatment.
>
> We find through subsequent experiments that when $T'=1$, LM-HT SNN can still maintain an accuracy of $81.0$%. We think that the above phenomenon might also hint that some sequence classification tasks can be effectively completed through global information on the time dimension.

---

> > ### Comment · Reviewer_wxgA · 2024-08-10
> >
> > I appreciate the author's response, which addresses my remaining concerns. I believe this is an interesting work for the neuromorphic community.

---

### Official Review · Reviewer_JrEi · 2024-07-10

**Soundness:** 3
**Presentation:** 2
**Contribution:** 3
**Rating:** 7
**Confidence:** 4

**Summary:**

This paper proposes a learnable multi-hierarchical threshold model for SNNs and call it LM-HT. The authors theoretically analyzes the equivalence between LM-HT, vanilla spiking models, and quantized ANNs, and demonstrate that LM-HT can achieve comparable performance comparable as quantized ANNs under a two-stage training-inference framework. The authors also introduce a hybrid training framework that improves the performance of converted SNNs within fewer time-steps.

**Strengths:**

* **Theoretical Analysis**: This paper provides a rigorous mathematical analysis of the equivalence between LM-HT and other models, offering a new perspective of SNNs and their learning capabilities.

* **Performance Improvement**: Experimental results demonstrate that LM-HT significantly outperforms previous SOTA methods on various datasets.

**Weaknesses:**

- **Limited Analysis of T-GIM**: While the paper introduces the Temporal-Global Information Matrix (T-GIM), it is unclear how it affects the learning process and performance of the LM-HT model.

- **Advantages of the LM-HT model**: While it is interesting to combine SNNs with quatization, it is unclear what advantages the proposed models have over traditional SNNs and quantized ANNs.

**Questions:**

* From Fig.1(d)-(e), it seems that the surrogate function used in this work is the Rectangle Surrogate Function. How about other types of surrogate functions?

**Limitations:**

This work can benefit from further discussion on the details of model parameters and method implementation.

---

> ### Author Rebuttal · Authors · 2024-08-05
>
> ## To Reviewer JrEi
> Thanks for your insightful and valuable feedback. We are glad that you found our work "provides a rigorous mathematical analysis", "offering a new perspective of SNNs" and "significantly outperforms previous SOTA methods". We would like to address your concerns and questions in the following.
>
> > 1. While the paper introduces the Temporal-Global Information Matrix (T-GIM), it is unclear how it affects the learning process and performance of the LM-HT model.
>
> Thanks for the question. From the perspective of forward propagation, as shown in Fig.2, we can observe that the T-GIM layer can encompass two training modes: vanilla STBP training (when $\Omega^l = \text{diag}(1,...,1)$) and quantized ANN training (when $\forall i,j\in[1,T], \omega_{ij}^l=\frac{1}{T}$). In the training stage of LM-HT SNN, we set the T-GIM layer parameters to be learnable, while combining the learnable leakage parameters $\lambda^l$ of the multi-threshold model, so that the LM-HT model can be dynamically optimized to any parameter state between vanilla STBP training and quantized ANN training.
>
> From the perspective of back-propagation, when we want to enhance the learning properties of LM-HT SNN simliar to that of quantized ANN, we can detach the temporal gradient term $\frac{\partial m^l(t)}{\partial m^l(t-1)}$, then we will have $\frac{\partial \mathcal{L}}{\partial W^l} = \sum_{t=1}^T \frac{\partial \mathcal{L}}{\partial m^l(t)} \sum_{j=1}^T \omega^l_{tj} s^{l-1}(j)\theta^{l-1}$.
>
> Here $\sum_{j=1}^T \omega^l_{tj} s^{l-1}(j)\theta^{l-1}$ denotes the comprehensive global information that extracted from different time windows; while $\frac{\partial \mathcal{L}}{\partial m^l(t)} = \sum_{j=1}^T \frac{\partial \mathcal{L}}{\partial m^{l+1}(j)} \omega^{l+1}_{jt} \theta^l {W^{l+1}}^{T} \frac{\partial s^l(t)}{\partial m^l(t)}$, which means that the gradients at each time-step is the comprehensive weighted result of the gradients at each time-step in the post-synaptic layer. At this point, we can observe that LM-HT SNN maintains computational equivalence with quantized ANN in both forward and backward propagation, while T-GIM layer plays an important regulatory role in it.
>
> > 2. While it is interesting to combine SNNs with quantization, it is unclear what advantages the proposed models have over traditional SNNs and quantized ANNs.
>
> Thanks for the question. The essence of the LM-HT model is to extract information from spike sequences in units of time windows with specific length, while the introduction of the T-GIM layer ensures that the LM-HT model can simultaneously focus on global information on the time dimension. In addition, the precondition that the input current is uniformly distributed within the time window has enabled the performance of LM-HT SNN to reach the same level as quantized ANNs.
>
> By comparison, vanilla spiking models can only extract local temporal information unidirectionally and have a certain degree of loss in learning accuracies, while quantized ANNs can only learn global average information on the time dimension. As shown in  Fig.2, we can note that the LM-HT model has the advantages of these two training modes and can effectively avoid the related disadvantages of them.
>
> > 3. From Fig.1(d)-(e), it seems that the surrogate function used in this work is the Rectangle Surrogate Function. How about other types of surrogate functions?
>
> Thanks for pointing it out. We have tried other types of surrogate functions before, but we tend to think that Rectangle Function is the most ideal choice. As the membrane potential of multi-threshold models before firing spikes actually represents comprehensive information within a time window, therefore, its surrogate gradients cannot be directly calculated by fitting Heaviside Function.
>
> For example, when $m^l(t)\in(k \theta^l, (k+1)\theta^l), k=0,...,L-1$, for vanilla surrogate functions (e.g. Sigmoid Function, Triangle Function) which aim to fit Heaviside Function, $\frac{\partial s^l(t)}{\partial m^l(t)} \rightarrow 0$, thereby leading to the gradient vanishing problem, although they can more richly reflect membrane potential information at each time-step. Therefore, we tend to think that this series of surrogate functions might be more suitable for vanilla single-threshold models.

---

> > ### Comment · Reviewer_JrEi · 2024-08-11
> >
> > Thanks for your rebuttal. I raise my score to 7.

---

### Official Review · Reviewer_zyi4 · 2024-07-10

**Soundness:** 4
**Presentation:** 3
**Contribution:** 4
**Rating:** 6
**Confidence:** 5

**Summary:**

This paper introduces the Learnable Multi-hierarchical Threshold (LM-HT) model, a novel approach to enhance the performance of SNNs to match that of ANNs. The LM-HT model dynamically adjusts the global input current and membrane potential leakage, and can be reparameterized into a standard single-threshold model for flexible deployment. The proposed model can integrate with ANN-SNN conversion framework to enhance the performance of converted SNNs under low time latency.

**Strengths:**

1. The LM-HT model provides a mathematical bridge between multi-threshold SNNs and quantized ANNs. The proposed reparameterization scheme allows for efficient hardware deployment.

2. The hybrid training framework based on the LM-HT model effectively addresses performance degradation issues in traditional ANN-SNN conversion methods.

3. Extensive experimental results demonstrate the superior performance across multiple datasets.

**Weaknesses:**

1. The author need to clarify the benefits of using the LM-HT model during the training phase and reparameterizing it to a vanilla single threshold model during the inference phase. For example, what are the advantages of this approach compared to directly training a single threshold model based on T-GIM modules?

2. Theorem 4.2 and Theorem 4.4 have demonstrated the equivalence of layer-by-layer output between the LM-HT model and quantized ANN during forward propagation, but the authors still need to analyze the equivalence of gradient calculation between the LM-HT model and quantized ANN from the perspective of back-propagation.

**Questions:**

According to Section 4.5 and Figure 3, it seems that the reparameterization procedure is simultaneously required for Conv&BN, T-GIM and neuron layers. Can the authors make a more detailed explanation about the parameter rectification for these three types of modules?

---

> ### Author Rebuttal · Authors · 2024-08-05
>
> ## To Reviewer zyi4
> Thanks for your constructive and valuable feedback. We are encouraged that you found our proposed method "effctive", "superior" and "provides a mathematical bridge between multi-threshold SNNs and quantized ANNs". We would like to address your concerns and questions in the following.
>
> > 1. What are the advantages of this approach compared to directly training a single threshold model based on T-GIM modules?
>
> Thanks for the question. Firstly, there is an advantage in terms of computational cost. During the training phase, compared to single-threshold models, using LM-HT models saves $L \times$ in both time and memory overhead ($O(LT)$ v.s. $O(T)$). Next is the enhancement of performance. The LM-HT model enhances the uniformity of spike sequences within each time window, combined with the gradient calculation mode of Eqs.(8)-(9), making LM-HT SNN have an equivalent training mode to quantized ANN, significantly improving its learning ability. Ultimately, our method can also reparameterize LM-HT SNN to vanilla SNN at no cost during the inference stage.
>
> > 2. The authors still need to analyze the equivalence of gradient calculation between the LM-HT model and quantized ANN from the perspective of back-propagation.
>
> Thanks for your comment. According to Eqs.(8)-(9), we can have
>
> $\frac{\partial \mathcal{L}}{\partial m^l(t)} = \sum_{j=1}^T \frac{\partial \mathcal{L}}{\partial m^{l+1}(j)} \omega^{l+1}_{jt} \theta^l {W^{l+1}}^{T} \frac{\partial s^l(t)}{\partial m^l(t)}$,
>
> and $\frac{\partial \mathcal{L}}{\partial W^l} = \sum_{t=1}^T \frac{\partial \mathcal{L}}{\partial m^l(t)} \sum_{j=1}^T \omega^l_{tj} s^{l-1}(j)\theta^{l-1}$.
>
> When $\forall i,j\in T, \omega^l_{ij}=\frac{1}{T}$ and $\forall t\in T, m^l(t)\in[\frac{\theta^l}{2}, (L+\frac{1}{2})\theta^l ]$, as $\frac{\partial s^l(t)}{\partial m^l(t)} = \text{sign}(\frac{\theta^l}{2} \leq m^l(t) \leq (L+\frac{1}{2})\theta^l )$, we will find that
>
> $\forall t\in [1,T], \frac{\partial \mathcal{L}}{\partial m^l(t)} = \frac{\theta^l }{T} \sum_{j=1}^T \frac{\partial \mathcal{L}}{\partial m^{l+1}(j)} {W^{l+1}}^{T}$,
>
> which means that $\frac{\partial \mathcal{L}}{\partial m^l(1)} = ... = \frac{\partial \mathcal{L}}{\partial m^l(T)}$.
>
> In additon, we can derive that $\frac{\partial \mathcal{L}}{\partial W^l} = \sum_{t=1}^T \frac{\partial \mathcal{L}}{\partial m^l(t)} r^{l-1}, r^{l-1} = \frac{1}{T} \sum_{j=1}^T s^{l-1}(j)\theta^{l-1}$, which is equivalent to the back-propagation mode of quantized ANN.
>
> > 3. Can the authors make a more detailed explanation about the parameter rectification for Conv&BN, T-GIM and neuron layers?
>
> Thanks for the question. For the bias terms in Conv&BN layers, due to the $L\times$ time-steps expansion in the inference stage, the addition operation at each time-step needs to be divided by $L\times$ (to ensure $I^l(t) + b^l_t = \sum_{j=L(t-1)+1}^{Lt} I^l_{IF}(j) + \hat{b^l_j}$, we set $\forall j\in [L(t-1)+1, Lt],  \hat{b^l_j} = b^l_t / L$).
>
> For T-GIM layers, due to the precondition that the multi-threshold model in the training stage is based on the uniform distribution of the input current within each time window, the values at each position within the $L\times L$ windows need to be divided by $L\times$ while expanding the shape of T-GIM in the inference stage (to ensure $\sum_{i=1}^T \omega^l_{ti} I^l(i) = \sum_{j=L(t-1)+1}^{Lt} \sum_{i=1}^{LT} \hat{\omega^l_{ji}} I^l_{IF}(i)$, we set $\forall i\in[1,T], \forall j\in [L(t-1)+1, Lt], \forall k\in [L(i-1)+1, Li], \hat{\omega^l_{jk}} = \omega^l_{ti} / L$).
>
> For spiking neural layers, the multi-threshold model during the training phase has no membrane potential leakage within each time window, and the leakage degree between windows can be directly copied from the leakage parameters $\lambda^l$ during the training phase ($\forall t \in [1,T], \forall j\in [L(t-1)+1, Lt), \hat{\lambda^l_j} = 1; \forall j=L,...LT,  \hat{\lambda^l_j} = \lambda^l_{j/L}$).

---

### Author Rebuttal · Authors · 2024-08-06

## To All Reviewers
Thanks for all your constructive and insightful feedbacks! Considering the concerns of some reviewers about detailed implementation of LM-HT SNN in the reparameterization process, we will provide an additional proof here regarding the computational equivalence before and after reparameterization:

### Computational Equivalence about the Reparameterization Process

**Theorem R.1** $\forall t, i\in[1,T], \forall j\in [L(t-1)+1, Lt], \forall k\in [L(i-1)+1, Li]$, when $s^{l-1}(t) = \sum_{j=L(t-1)+1}^{Lt} s_{IF}^{l-1}(j)$, if $\hat{b^l_j} = b^l_t / L, \hat{\omega^l_{jk}} = \omega^l_{ti} / L$, we will have $I^l(t) = \sum_{j=L(t-1)+1}^{Lt} I_{IF}^l(j)$. Here $\hat{b^l}, \hat{\Omega^l}$ denote the rectified bias term and T-GIM layer after the reparameterization process.

*proof* Firstly, it is obvious that $I^l(t), I_{IF}^l(j)$ can be rewritten as $I^l(t) = \sum_{i=1}^T \omega^l_{ti} (W^l s^{l-1}(i) + b^l_i)$ and $I_{IF}^l(j) = \sum_{i=1}^{LT} \hat{\omega^l_{ji}} (W^l s_{IF}^{l-1}(i) + \hat{b^l_i})$.

Considering the precondition $\hat{b^l_j} = b^l_t / L, \hat{\omega^l_{jk}} = \omega^l_{ti} / L$, we will have:

$I_{IF}^l(j) = \sum_{i=1}^{T} \sum_{k=L(i-1)+1}^{Li} \hat{\omega^l_{jk}} (W^l s_{IF}^{l-1}(k) + \hat{b^l_k}) $

$= \sum_{i=1}^{T} \frac{\omega^l_{ti}}{L} \sum_{k=L(i-1)+1}^{Li} (W^l s_{IF}^{l-1}(k) + \frac{b^l_i}{L})$.

Then we can further have:

$ \sum_{j=L(t-1)+1}^{Lt} I_{IF}^l(j) = \sum_{j=L(t-1)+1}^{Lt} \sum_{i=1}^{T} \frac{\omega^l_{ti}}{L} \sum_{k=L(i-1)+1}^{Li} (W^l s_{IF}^{l-1}(k) + \frac{b^l_i}{L}) $

$= \sum_{i=1}^{T} \frac{\omega^l_{ti}}{L} \sum_{j=L(t-1)+1}^{Lt} (W^l \sum_{k=L(i-1)+1}^{Li}  s_{IF}^{l-1}(k) + b^l_i) $

$= \sum_{i=1}^{T} \frac{\omega^l_{ti}}{L} \sum_{j=L(t-1)+1}^{Lt} W^l (s^{l-1}(i) + b^l_i) $

$= \sum_{i=1}^T \omega^l_{ti} (W^l s^{l-1}(i) + b^l_i)$

$=I^l(t)$.

Due to the fact that the calculation process of the spike sequences passing through Conv&BN and T-GIM layers can be abstractly described by Theorem R.1, we can conclude that the sum of the input currents within the corresponding time windows before and after reparameterization remains unchanged. The spike sequences obtained by passing the input currents through the spiking neuron layer will also satisfy the precondition of Theorem R1 ($s^{l}(t) = \sum_{j=L(t-1)+1}^{Lt} s_{IF}^{l}(j)$). Therefore, we can prove the computational equivalence before and after the reparameterization process.

---

### Decision · Program_Chairs · 2024-09-25

**Decision:**

Accept (poster)

**Comment:**

The reviewers concurred that this work (1) has solid theoretical analysis and empirical validation, and (2) will be of interest to the NeurIPS community. The authors' reply to the reviewers comments helped strengthen the paper's position, both on the theory side (by showing additional derivations and providing justification of the method's usefulness) and the empirical side (by providing further experimental validation and better statistics).